# Relationship between Child Abuse and Delinquent Behavior in Male Adolescents Deprived of Liberty

**DOI:** 10.3390/ijerph192416666

**Published:** 2022-12-12

**Authors:** Patricio Alfredo Vallejo Valdivieso, Graciela Hernestina Zambrano Pincay, Cristina M. Beltran-Aroca, Eloy Girela-Lopez

**Affiliations:** 1Department of Biological Sciences, Faculty of Health Sciences, Medicine Career, Universidad Técnica de Manabí, Portoviejo 130105, Ecuador; 2Section of Legal and Forensic Medicine, Facultad de Medicina y Enfermería, Universidad de Cordoba, 14004 Cordoba, Spain

**Keywords:** child abuse, risk factors, criminal behavior, adolescents deprived of liberty

## Abstract

The objective is to identify the prevalence of child abuse and criminal behavior among young male prisoners in Ecuador. Method: A total of 425 young people between 12 and 18 years of age, deprived of liberty from different centers for adolescent offenders in Ecuador, were used. The level of abuse to which they had been subjected in childhood was evaluated, as well as the risk factors present in their history. The relationship between abuse, risk factors, and criminal behavior was analyzed. Results: A high prevalence of the different types of abuse was found in the mean age of 15.03 years standard deviation (SD = 1.34). Conclusion: In addition, a relationship was discovered between the abuse suffered during childhood and the risk factors present in criminal behavior during adolescence, which was demonstrated in this article. The practical implications of these results are discussed, taking into consideration their relevance for prevention.

## 1. Introduction

Child abuse according to the proposal included in the world situation report on the prevention of violence by the World Health Organization [1] can be defined as the abuse and neglect of children under 18 years of age, including here all the types of physical or emotional abuse, sexual abuse, negligence, and commercial or other exploitation that result in actual or potential harm to the child’s health, survival, development, or dignity in the context of a relationship of responsibility, trust or power.

Thus, under a broad conception of child abuse, we can affirm that this term is used to describe abusive or negligent acts perpetrated by adults or older youth against children. [2]. The problem is associated with negative results, including antisocial behavior. In Ecuador, there are few studies on the subject due to the difficulties in obtaining information, unlike Western/American countries that have a different penal system, since regulatory laws in certain cases submit the minor to a trial in which he will be punished as an adult.

Regarding the types of child abuse, different classifications have been developed. One of the most widely used is the ‘hierarchical classification system’ [3] which distinguishes physical abuse, emotional abuse, child neglect, and sexual abuse. Physical abuse is understood as those acts that involve physical punishment and aggression that can injure a minor and even end her/his life. Emotional abuse, also called psychological abuse, can be defined as abuse related to lack of care by the caregiver, causing the child to lack support and remain in an environment that is unsuitable for their development [1]. Regarding negligence, this has been divided into two categories: physical, understood as a lack of attention to the basic needs of minors; and the psychological, more related to the lack of attention to emotional needs [4].

Regarding the prevalence of child abuse, it is generally difficult to obtain reliable data, especially when it occurs in the family context [5]. Apart from this difficulty, the literature reflects a high prevalence of child abuse, exceeding 50% in Africa, Asia, and North America, and 30% in Latin America [6].

Specifically, in America, it has been seen that these figures vary depending on the level of poverty, ranging between 25% and 40% [1]. Differentiating the typologies already mentioned, there is no consensus about which is the most prevalent. Some studies point to psychological abuse at (36.3%), followed by physical (22.6%), emotional (18.4%), and physical (16.3%) neglect [7]. However, other studies indicate that neglect is the most frequent type of abuse (41.5%), followed by physical abuse (28.4%) [8]. In this sense, it must be taken into account that there is usually more than one form of abuse at the same time [9].

According to the World Health Organization [1], children and adolescents constitute a population group that is especially vulnerable to interpersonal violence, highlighting the relevance of the consequences of the experience of abuse for children, which interfere in all areas of development. As refered by [10], this abuse “is related to processes of maladjustment during development and to patterns of behavioral and emotional problems, which are acquired in the environment in which the minor develops, projecting in the future behaviors of coping that may culminate in criminal acts”.

In this sense, Ref. [11] considers that it is difficult to specify the consequences of a traumatic event, since it implies various causes, and these appear over the years. However, it specifies that a minor who has been exposed to experiences of abuse in childhood has a greater risk of presenting problems in interpersonal relationships such as mistrust and avoidance of private and social relationships. According to [12], minors who have suffered some type of abuse (neglect, physical abuse, emotional abuse, sexual abuse) are more likely to present alterations in all areas of their development (emotional, behavioral, social, academic, and physical health) than those minors who have not suffered this situation. On the other hand [13], the presence of disruptive or antisocial behaviors during adolescence are interrelated with childhood abuse. In this vein, minors who have suffered or are suffering abuse are more likely to engage in externalizing behaviors such as school absenteeism, running away, crimes against private property, physical fights, use of weapons, as well as risky sexual behaviors [14].

In this way, it should be noted that, during upbringing, the family plays a fundamental role, since it constitutes an exogenous factor of great relevance for the development of antisocial behaviors [15]. Thus, it plays a key role in the eradication or proliferation of habits concomitant with the development of a personality related to criminal behavior in adolescents [16]. Similarly, it has been found that abuse is highly prevalent in poor and marginalized sectors of the population and constitutes a significant threat to the healthy development of children who grow up in these environments [11]. It is pertinent to have public policies where the State and the family work towards promoting a crime-free society [15]. In this sense, understanding the magnitude and severity of child abuse is essential to develop clinical interventions and social policies in order to protect children at risk and treat those who have already been victimized [17].

There is a need to study the relationship between the experience of abuse and criminal behavior in adolescence. It is necessary to point out that, in the Republic of Ecuador, adolescent offenders conform to a precise legal category, an adolescent being within the range of 12 to 17 years, 11 months, and 29 days. Children under 12 years of age are exempt from the measure as of the enactment of the Code for Children and Adolescents [18]. Adolescents who have been declared criminally responsible, in accordance with the provisions of article 369 of the special legal text, will have socio-educational measures imposed, which are defined as actions ordered by the judicial authority when the responsibility of the adolescent has been declared in an act typified as a criminal offence. Its purpose is to achieve the social integration of the adolescent and the repair or compensation of the psycho-emotional damage caused. In Ecuador, there are eleven detection centers for adolescents, where robbery, murder, rape, and drug trafficking are the four most common crimes [19]. However, obtaining data from adolescents deprived of liberty is a great challenge, due to the confidentiality criteria associated with the postulates of the Doctrine of Comprehensive Protection [20], so there is little literature on the matter.

Based on what has been described, the objective is to identify the prevalence of child abuse and criminal behavior among young male prisoners who are involved in justice in Ecuador, for which the question arises: Do you consider that the prevalence of abuse is related to criminal behavior in adolescents in the study?

Although it is not possible to establish a causal relationship between child abuse and criminal behavior, there are some basic relationships between the two phenomena. It has been observed that children physically and/or emotionally assaulted would be more prone to commit aggressive criminal acts, and that children who suffered from neglect would be more likely to commit crimes against property. In addition, it has been pointed out that a significant percentage of the criminal population has a history of violence in childhood.

Families that live in violence are in a situation of isolation, rejection, and negative feelings, among others, regarding their community. This aspect has been considered as an element associated with the generation of criminal behavior within the group, increasing the biopsychosocial imbalance of adolescents.

It is important to keep in mind that the legal norms in force in the Ecuadorian State make it difficult to address other aspects that this issue requires, such as monitoring the reintegration into society once the sentence has been served or monitoring the effects of child abuse to continue committing any type of crime; this is not allowed since, without authorization to carry it out, article 5 of the Comprehensive Organic Criminal Code [21] that refers to procedural principles would be violated.

Based on the Code of Childhood and Adolescence in its Article 50, in relation to the “right to personal, physical, psychological, cultural, affective and sexual integrity”, together with Article 54 that refers to the “right to the reservation of the information on criminal records”, together with Article 51 Numeral 7 that emphasizes that “persons deprived of liberty are recognized… to have protection measures for girls, boys, adolescents”, which motivated the competent authorities not to provide information on female adolescents deprived of liberty.

The Code for Children and Adolescents in Article 306 mentions that “adolescents who commit offenses classified in the Comprehensive Organic Criminal Code will be subject to socio-educational measures for their responsibility in accordance with the precepts of this Code”.

This same legal body, in its Article 385, establishes the “application of socio-educational measures in crimes sanctioned in the Comprehensive Organic Criminal Code, where in its Numeral 3 it refers to” cases of crimes sanctioned with a custodial sentence of more than ten years where the measure of reprimand and institutional internment from four to eight years will be applied. Thus, Article 388 emphasizes that “the adolescent sentenced upon reaching the age of majority will continue with the imposed socio-educational measure. If it is a deprivation of liberty socio-educational measure, it will remain in a special section in the same Center for adolescent offenders.”

To achieve the objective and answer the question raised, the Constitutional rights and guarantees were respected together with the Code for Children and Adolescents, for which we reserve the right to the confidentiality of the information.

## 2. Method

### 2.1. Study Design

A study of a quantitative nature was proposed, under a non-experimental cross-sectional design [22]. The level was relational since it sought to identify the factors of parental child abuse linked to criminal behavior in male adolescents deprived of liberty in Ecuador.

### 2.2. Participants

A sample of 425 files was obtained (see Table 1) of male adolescents with a mean age of 15.03 years (SD = 1.34), accused of different crimes: robbery (n = 289), drugs (n = 72), rape (n = 43), and murder (n = 21). The reviewed files correspond to the period from November 2018 to January 2020. The selection of the participants was carried out through a non-probabilistic sampling method. For legal reasons, it was not possible to access all the files. The inclusion criteria applied were the following: (a) that the age was between 14 and 17 years, 11 months, and 29 days; (b) that he had a final sentence or was in court for prosecution of a crime typified in Ecuadorian law; and (c) that the file registered in the CAI confirmed having suffered child abuse in the sample taken in the offending centers.

### 2.3. Measuring Instruments

To assess the intensity of the abuse suffered, the Child Abuse Detection Scale [23] was used. This scale is made up of 23 items, which the adolescent has to answer based on a 5-point Likert-type scale, from never (1) to always (5). These responses are categorized into three levels: (1) lower intensity, corresponding to the scores never (1) and very rarely (2); (2) moderate, corresponding to the score a few times (3); and (3) greater intensity, corresponding to the scores often (4) and always (5). The scale is divided into four factors (see Table 2), corresponding to the four types of abuse. It has a good internal consistency index (α = 0.788).

In addition, a checklist was used, an observation-based instrument proposed by [24].

This list focuses on the risk factors, so that it allows the presence or absence of three areas or contexts of risk to be indicated for juvenile delinquency from 11 items, which the evaluator responds to in a dichotomous way (presence vs. absence) based on the child’s file. The contexts that it explores are: (a) marginal, constituted by the presence of marginalization in the youth’s environment and/or lack of basic services, such as a family housing area; (b) family, made up of elements associated with the presence of moral or economic support from the parents to the young person; and (c) educational, related to learning problems and elements related to schooling, such as not being in the course corresponding to their age or not being enrolled in a school grade. This scale was evaluated through expert judgments and was found to be valid.

### 2.4. Procedure

The researcher went to each of the centers for adolescent offenders to review the files and evaluate the presence of risk contexts for juvenile delinquency. For those who met the criteria, the following instruments were applied: the child abuse scale and the observational checklist. To do this, all participants were informed of the content of the study and informed consent was requested to participate, without the need for the express authorization of those responsible, according to the provisions of the [18], which recognizes them as subjects of rights based on the progressive exercise of their rights. In this way, participation was voluntary, and the participants agreed both with the completion of the questionnaire and with the review of their files.

### 2.5. Ethics

This research was approved by the Bioethics Committee of Universidad Técnica de Manabí and is found in Volume: 018-04 Folio: 18-04-3. I also declare that the informed consent was presented verbally and in writing.

This study focuses on the elements of child maltreatment, including: (1) emotional neglect, (2) physical maltreatment, (3) emotional maltreatment, and (4) physical neglect, along with behavioral disorders such as theft, drugs, rape, and murder, framed in marginal, family, and educational contexts.

### 2.6. Analysis of Data

The data were descriptively analyzed in order to determine the absolute and percentage frequency distributions. On the other hand, a comparison of means was carried out in the type of abuse according to the context (marginal, family, and educational) in each risk context using Student’s *t*-test for independent samples. In order to find out the differences between the types of crimes according to the type of abuse, a one-way ANOVA parametric analysis was performed, and post hoc tests with HSD Tukey were used to make multiple comparisons. Finally, the size of the effect or size of the differences between two groups is quantified in order to evaluate the magnitude of the result and to know the scope of the findings.

## 3. Results

### 3.1. Descriptive Analysis

The study participants were made up of 425 male adolescents deprived of liberty for various crimes reported in the files consulted. Table 3 shows that the most frequent crimes were robbery 68% (n = 289), drugs 17% (n = 72), rape 10% (n = 43), and murder 5% (n = 21). However, the majority of young people are exposed to negative elements in their marginal, family, and educational contexts.

Table 4 was constructed with the sum of the scores collected in the measurement instrument from the responses issued by the adolescents. It can be seen that the types of abuse exhibit similar behavior in the reported descriptions, that is, around the central values of the 5-point Likert scale, from never (1) to always (5).

Table 5 shows the percentage distribution of the types of child abuse according to levels of presence in the adolescents studied. Significant percentages are observed at moderate and high levels, which points to the presence of victimization in young people. Indeed, in Table 5, it can be read that physical and emotional abuse and physical neglect, in that order, have a moderate presence.

Table 6 reports, descriptively, the presence or absence of exposure to negative elements of young detainees within three contexts. It is observed in Table 6 that the family context (precarious economic situation and family abuse) presents a higher percentage differentiating from the rest, followed by the educational one (school dropouts) and, finally, the marginal one (drug use and living in marginalized and inadequate environments), which indicates that it favors the criminal behavior of the minor.

### 3.2. Comparisons between the Types of Child Maltreatment and Context

Table 7 shows the comparisons between the type of child abuse and the context. Regarding the marginal context, the young people who were exposed to a marginal context in negative circumstances appeared to have higher averages in the types of abuse studied. These differences were statistically significant (last three columns) and had a medium effect size. In other words, the young people who experienced negative marginal situations also experienced greater emotional abandonment, physical abuse, and affective and physical abandonment. Regarding the family and educational context, very similar results to the previous ones are observed. The adolescents who reported a family and educational context with negative elements also appeared to have a higher average in the types of abuse.

### 3.3. Comparisons between Types of Crimes and Types of Child Maltreatment

Table 8 reports statistically significant differences with a large effect size for each type of child abuse and type of crime and, indeed, between some of the means of the scores of the variables of the type of abuse according to the type of crime. An analysis was carried out post-hoc by Tukey to find, specifically, among whom this difference occurs.

Table 9 shows the results of all multiple comparisons using Tukey’s post-hoc analysis. For the characteristics of emotional abandonment, there are statistically significant differences between all the reported combinations, except Robo-Drug, with a mean difference of 0.776 (*p* = 0.442). In addition, the size of the effect between the differences found can be classified as “large variations”. For falls involving physical abuse, there are statistically significant differences between the types of crimes, less so for rape and murder. Moreover, the size of the effect between the differences found is considered large.

In the emotional abuse scores, there are statistically significant differences between the types of crimes, except for drug–murder. The effect size of the differences found is considered large. Finally, for the physical abandonment scores, statistically significant differences were found between the types of crimes, except for robbery–drugs. The effect size of the differences found is considered large.

In a disaggregated way, the relationship between the objective, the question raised, and the discussion between crimes committed and types of abuse suffered will be analyzed.

For the typology of affective and support needs, differences were found between the types of crime (see Table 10). On the other hand, the a posteriori contrasts, with the Bonferroni level of protection (*p* = 0.05/6 = 0.003), revealed the differences between the groups. Thus, those adolescents who had committed robberies reported having suffered less abuse of this type than adolescents who had committed drug-related crimes (d = 4.66), rape (d = 2.46), and murder (d = 0.33). Likewise, adolescents who had committed drug-related offenses reported having suffered more abuse of this type than those who had committed rape (d = 1.97) and murders (d = 5.13). Adolescents who had committed rape had higher scores than those who had committed murder (d = 1.98).

Regarding the physical abuse suffered during childhood, the results showed that there are differences depending on the type of crime committed (see Table 10). On the other hand, the a posteriori contrasts, with the Bonferroni level of protection (*p* = 0.05/6 = 0.003), revealed the differences between the groups. Thus, those adolescents who had committed robberies reported having suffered less abuse of this type than adolescents who had committed drug-related crimes (d = 4.93), rape (d = 1.71) and murder (d = 2.31). Likewise, adolescents who had committed drug-related offenses reported having suffered more abuse of this type than those young people who had committed rape (d = 2.92) and murders (d = 2.30). Adolescents who had committed murder had higher scores than those who had committed rape (d = 0.43).

For the emotional abuse suffered, the results show that there are differences depending on the type of crime carried out (see Table 10). On the other hand, the a posteriori contrasts, with the Bonferroni level of protection (*p* = 0.05/6 = 0.003), revealed the differences between the groups. Thus, those adolescents who had committed robberies reported having suffered less abuse of this type than adolescents who had committed drug-related crimes (d = 4.48), rape (d = 2.03), and murder (d = 0.61). Likewise, adolescents who had committed drug-related offenses reported having suffered more abuse of this type than those young people who had committed rape (d = 2.13) and murders (d = 0.81). Adolescents who had committed murder had higher scores than those who had committed rape (d = 0.93).

With regard to physical abandonment suffered during childhood, the results show differences depending on the type of crime committed. On the other hand, the a posteriori contrasts, with the Bonferroni level of protection (*p* = 0.05/6 = 0.003), revealed the differences between the groups. Thus, those adolescents who had committed robberies reported having suffered less abuse of this type than adolescents who had committed drug-related crimes (d = 3.88), rape (d = 1.78), and murder (d = 0.38). Likewise, adolescents who had committed drug-related offenses reported having suffered more abuse of this type than those young people who had committed rape (d = 2.66) and murders (d = 3.83). Finally, adolescents who had committed rape had higher scores than those who had committed murder (d = 1.15).

## 4. Discussion

Child abuse is defined as the abandonment of children under the age of 18, the result of emotional abandonment caused by a lack of attention as their affective and relational needs are not met. Abuse is the result of injury or danger; emotional abuse is inappropriate and disrespectful treatment that affects relationships with other people; physical abandonment of a minor caused by inadequate and insecure supervision can be one of the causes of criminal acts.

In this study, the interest is focused on identifying the factors of child abuse linked to criminal behavior in male adolescents deprived of liberty in Ecuador. Compared to other international studies [23,24,25,26], the sample size is considered high for similar research on the issues of abuse and juvenile delinquency, among others.

In general terms, regarding child abuse, it was found that the adolescents surveyed were victims, at different levels, of parental actions constituting child abuse. Likewise, the presence of risk contexts was verified through the files of the young people. However, when comparing each factor associated with child abuse and risk contexts, the presence of negative situations linked to family support, learning problems, and marginalized environments were highly differentiating characteristics among adolescents who suggested child abuse (moderate to high). This result corresponds to the findings of [27], for whom social and family deficiencies were notorious in all the young people studied and, within their conditions, the presence of conditions of family marginalization and the like.

In general, studies indicate the existence of aspects that contribute to minimizing the probability of committing crimes, such as living with parents [27]. This suggests that the same lack of family unity, communication with parents, and problematic situations within the family nucleus increase the risk of committing a crime. Similarly, having access to education helps to reduce crime because young people have less leisure time [24]. Our results agree with the literature, given that the presence of school dropouts among adolescents was high. The marginality of the family and the environment are characteristics that tend to encourage delinquent behavior in adolescents [26]. In this respect, our results are in line with what has been found in other investigations.

With regard to the relationship between the types of child abuse and the type of crime committed, in the young people who committed drug crimes, there was a high and statistically significant intensity in the four types of abuse studied. In addition, this group of young people had a higher risk of committing crimes because characteristics associated with little family support, school abandonment, and living in marginal areas were discovered in their files. Regarding the crime of murder, a high intensity of physical and emotional abuse was found, while the factors of affective and support needs and physical abandonment were found to have a low intensity. Likewise, it was marked by the presence of little family support and school desertion. In this sense, marginalization and family deficiencies are common among all young people accused of crimes. For the crime of rape, a moderate intensity of abuse is observed for the four typologies studied. Regarding risk contexts, the literature indicates that the very activity of young people, family dysfunction, and cohabitation in marginalized areas lead them to have a greater risk of being involved in a criminal act [24]. However, our results do not show a clear trend in this regard. Regarding the crime of theft, adolescents have a low score in all types of abuse. In addition, young people who commit this type of crime have the lowest percentages in the three risk contexts studied.

In general terms, it can be concluded that the presence of child abuse, specifically affective and attachment needs, physical abuse, and emotional abuse, is related to delinquent behaviors, in accordance with previous literature [28]. Similarly, marginal condition, lack of family support, and early school dropout all contribute to high rates of psycho-emotional instability in young people, which can lead to them seeking refuge in drug use and joining gangs, eventually becoming juvenile delinquents [29]. Given this [30], consider the following confounding factors in the development of delinquent behavior in adolescents. Thus, they point out that the adolescent, seeing himself without economic support and with few options for school improvement due to insufficient family support, can be frustrated and socially excluded, assuming inappropriate behaviors that contribute to the triggering of actions outside the law.

The limitations of the present study should be noted. Thus, the sample we have is exclusively male, so the generalization of the results is limited. Regarding the collection of information, the reference to the typologies of child abuse has been obtained through a self-reported questionnaire, which can give rise to biases; while the information related to the risk contexts has been obtained from what is indicated in the files, so it is limited by the rigor with which the information was added at the time. It should be noted that this is a cross-sectional study, so it only allows us to establish correlations.

Thus, for future studies, it is necessary to also explore these variables in women, given that the literature indicates differences between the sexes [31]. Likewise, in order to know more precisely the relationship between abuse and delinquency, it would be interesting to carry out an investigation with a longitudinal design. This would allow for establishing causal relationships, as well as measuring the variables of child abuse and risk contexts at the time they occur. It would be interesting to study possible moderating variables, such as the frequency or chronicity of the abuse suffered [32]. In general, we can conclude that the results obtained support the relationship between abuse in childhood and criminal behavior in the adolescent stage. Likewise, the relevance of the different risk contexts in the development of these behaviors is verified. All this points, once again, to the great importance of the protection of minors against the different forms of abuse, as well as the intervention of those who have been victims of it.

## 5. Conclusions

Physical and emotional abuse are the main generators of criminal behavior in adolescents deprived of liberty, a situation that leads to a health problem that should set off the alarms of the respective entities of the Public Power, especially the actors that make up the Ministry of Health, who must enable a comprehensive socio-psycho-pedagogical care model for the treatment of victims, particularly the child and adolescent population that has been declared as especially vulnerable subjects.

It stands out from the study that the young people were victims of physical-emotional abuse in their childhood, which brought serious consequences for their lives, among them, having incurred in the commission of infringing behaviors whose substrate is violence, which warranted the application of the most burdensome socio-educational measure of the Juvenile Penal System, such as deprivation of liberty.

In the same way, all the means and measures available to the Comprehensive Protection System must be reinforced, to provide the best care to those who have been victims of abuse, in any of its modalities, understanding that the effects of said practices are generated in the medium and long-term, through behaviors that challenge social norms.

## Figures and Tables

**Table 1 ijerph-19-16666-t001:** Center of origin of the adolescents.

Center of Origin	Fr	%
Ambato	45	10.6%
Esmeraldas	71	16.7%
Guayaquil	118	27.8%
Ibarra	65	15.3%
Machala	30	7.1%
Quito	96	22.6%
Total	425	100.0%

**Table 2 ijerph-19-16666-t002:** Factors of child abuse.

Factor	Description
Affective and support needs (emotional abandonment)	These items are scored in reverse. When high scores are obtained, it indicates emotional abandonment; that is, that the parents are not aware of the emotional, social, and educational needs of the minor.
Physical abuse	The high score in this factor refers to physical violence towards the minor.
Emotional abuse	High scores refer to the denigration of the minor through unpleasant phrases, insults, or other verbalizations that may cause him harm.
Physical neglect	The high scores refer to the parents’ lack of attention to the child’s hygiene and body care, as well as to their basic needs, such as food and rest.

**Table 3 ijerph-19-16666-t003:** Descriptive statistics for the type of crime and context (n = 425).

Variables	Frequency	Percentage
Type of Crime		
Robery	289	68%
Drug	72	17%
Violation	43	10%
Murder	21	5%
Fringe context ^a^		
Present	224	53%
Absent	201	47%
Family context ^b^		
Present	245	58%
Absent	180	42%
Educational context ^c^		
Present	229	54%
Absent	196	46%
Total	425	100%

Note: ^a^ urban marginalization and presence of drugs; ^b^ precarious economic situation and behaviors of domestic violence; ^c^ learning problems and youth dropout.

**Table 4 ijerph-19-16666-t004:** Descriptive statistics for the types of abuse (n = 425).

Variables	Min	Max	M	SD	Asymmetry	Kurtosis
Emotional abandonment	7	21	11.59	4.66	0.913	−0.605
Physical abuse	6	21	12.13	4.11	0.821	−0.632
Emotional abuse	6	24	12.53	4.21	0.751	−0.616
Physical neglect	6	18	11.60	3.66	0.571	−0.857

Note: Min = Minimum; Max = Maximum; M = Mean; SD = Standard deviation.

**Table 5 ijerph-19-16666-t005:** Distribution of the sample according to the level and type of child abuse suffered.

	Under	Moderate	High
Emotional abandonment	150 (35.3%)	152 (25.8%)	123 (28.9%)
Physical abuse	46 (10.8%)	236 (55.5%)	143 (33.6%)
Emotional abuse	82 (19.3%)	201 (47.3%)	142 (33.4%)
Physical neglect	106(24.9%)	192 (45.2%)	127 (29.9%)

**Table 6 ijerph-19-16666-t006:** Distribution of the sample according to the context for juvenile delinquency.

	Present	Absent
Fringe Context ^a^	224 (52.7%)	201 (47.3%)
Family Context ^b^	245 (57.6%)	180 (42.4%)
Educational Context ^c^	229 (53.9%)	196 (46.1%)

Note: ^a^ urban marginalization and presence of drugs; ^b^ precarious economic situation and behaviors of domestic violence; ^c^ learning problems and youth dropout.

**Table 7 ijerph-19-16666-t007:** Comparison of means for the type of abuse according to the context.

		N	M	DT	t (gl)	*p*	d
**Fringe Context ^a^**							
Emotional abandonment	Present	224	13.52	5.14	7.42 (423)	0.000	0.74
	Absent	201	10.18	3.68			
Physical abuse	Present	224	13.92	4.40	7.93 (423)	0.000	0.79
	Absent	201	10.82	3.33			
Emotional abuse	Present	224	14.25	4.37	7.46 (423)	0.000	0.74
	Absent	201	11.27	3.61			
Physical neglect	Present	224	13.12	3.89	7.60 (423)	0.000	0.75
	Absent	201	10.47	3.03			
**Family Context ^b^**							
Emotional abandonment	Present	245	12.98	5.06	5.79 (423)	0.000	0.56
	Absent	180	10.41	3.92			
Physical abuse	Present	245	13.48	4.34	6.47 (423)	0.000	0.62
	Absent	180	10.97	3.52			
Emotional abuse	Present	245	13.72	4.43	5.48 (423)	0.000	0.53
	Absent	180	11.52	3.73			
Physical neglect	Present	245	12.73	3.91	6.06 (423)	0.000	0.59
	Absent	180	10.62	3.12			
**Educational Context ^c^**							
Emotional abandonment	Present	229	12.80	5.05	5.13 (423)	0.000	0.50
	Absent	196	10.51	3.98			
Physical abuse	Present	229	13.28	4.37	5.62 (423)	0.000	0.54
	Absent	196	11.09	3.57			
Emotional abuse	Present	229	13.53	4.40	4.69 (423)	0.000	0.46
	Absent	196	11.64	3.82			
Physical neglect	Present	229	12.48	3.96	4.78 (423)	0.000	0.46
	Absent	196	10.80	3.17			

Note: ^a^ urban marginalization and presence of drugs; ^b^ precarious economic situation and behaviors of domestic violence; ^c^ learning problems and youth dropout. The t-statistic (df) is the result of the Student’s *t*-test with 423 degrees of freedom. One-way Anova and d represent the size based on differences between groups.

**Table 8 ijerph-19-16666-t008:** ANOVA results for types of child maltreatment and types of crimes.

	Type of Crimes						
Types of Abuse	Stole (*n* = 289)	Drug (*n* = 72)	Violation (*n* = 43)	Murder (*n* = 21)			
	M (DE)	M (DE)	M (DE)	M (DE)	F(3, 421)	*p*	η^2^
Emotional abandonment	9.2 (2.3)	19.7 (1.9)	15.1 (2.8)	9.6 (1.9)	440.00	0.000	0.785
Physical abuse	9.3 (2.0)	19.3 (1.3)	13.6 (2.7)	14.9 (3.3)	410.52	0.000	0.745
Emotional abuse	10.2 (2.1)	19.1 (1.2)	14.8 (2.9)	17.8 (3.8)	373.86	0.000	0.727
Physical neglect	9.8 (2.2)	17.7 (0.2)	13.7 (2.1)	10.7 (3.4)	282.16	0.000	0.668

Note: The F statistic is the result of the one-way Anova and η^2^ represents the effect size based on the explained variance.

**Table 9 ijerph-19-16666-t009:** Multiple comparisons between types of child abuse and types of crime.

Multiple Comparisons	Mean Differences	DE	*p*	d
Emotional Abandonment				
Robbery–Murder	10.49	0.303	0.000	4.669
Robbery–Rape	5.940	0.376	0.000	2.475
Robo-Drug	0.776	0.519	0.442	0.337
Drug–Rape	4.551	0.443	0.000	1.987
Drug–Murder	9.714	0.570	0.000	5.157
Rape–Murder	5.164	0.612	0.000	2.005
Physical abuse				
Robbery–Murder	9.361	0.274	0.000	4.935
Robbery–Rape	3.651	0.340	0.000	1.717
Robo-Drug	4.926	0.470	0.000	2.311
Drug–Rape	5.710	0.401	0.000	2.943
Drug–Murder	4.435	0.516	0.000	2.317
Rape–Murder	1.276	0.554	0.099	0.435
Emotional abuse				
Robbery–Murder	8.879	0.291	0.000	4.494
Robbery–Rape	4.577	0.361	0.000	2.037
Robo-Drug	7.619	0.499	0.000	3.347
Drug–Rape	4.302	0.425	0.000	2.145
Drug–Murder	1.260	0.547	0.099	0.613
Rape–Murder	3.042	0.587	0.000	0.940
Physical neglect				
Robbery–Murder	7.857	0.279	0.000	3.894
Robbery–Rape	3.921	0.346	0.000	1.787
Robo-Drug	0.891	0.478	0.246	0.386
Drug–Rape	3.936	0.408	0.000	2.669
Drug–Murder	6.966	0.525	0.000	3.863
Rape–Murder	3.030	0.563	0.000	1.161

Note: d represents size based on differences between groups.

**Table 10 ijerph-19-16666-t010:** Mean differences in the type of abuse for the committed crime factor (robbery, drug, rape, murder).

	N	M	DT	F	*p*
**Affective and Support Needs**					
Robbery	289	9.18	2.33	440	0.000
Drug	72	19.67	1.88		
Violation	43	15.12	2.85		
Murder	21	9.95	1.88		
**Physical Abuse**					
Robbery	289	9.93	2.02	373	0.000
Drug	72	19.29	1.26		
Violation	43	13.58	2.73		
Murder	21	14.86	3.32		
**Emotional Abuse**					
Robbery	289	10.19	2.13	282	0.000
Drug	72	19.07	1.16		
Violation	43	14.77	2.93		
Murder	21	17.81	3.80		
**Physical Neglect**					
Robbery	289	9.82	2.21	282	0.000
Drug	72	17.68	0.92		
Violación	43	13.74	2.10		
Murder	21	10.71	3.44		

Note: gl (3, 421).

## Data Availability

https://drive.google.com/drive/folders/1rHnc9CJH1pgMlWbyLR3o9ZGLcdHu-kGf?usp=sharing, Uploaded to the drive on 15 October 2022.

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
