# Peer review of "Relationship between Child Abuse and Delinquent Behavior in Male Adolescents Deprived of Liberty"

_ijerph, 2022, doi:10.3390/ijerph192416666_

Round 1

Reviewer 1 Report

Dear authors,

First of all, I would like to say that the subject of the study is very interesting and necessary. And secondly, I am aware of the work behind this article. However, in my opinion, it needs to be improved in order to be published. I consider that this version of the article is not sufficient for a Q1 journal.

I advise you to look at other articles to improve yours. The introduction is a bit messy and some content is not related to the article (eating disorders). There are also formatting errors and missing references. In the methodology, sections are missing (or appear at the end of the article). The objectives and hypotheses should be further developed. and finally, the results should be linked to the objectives, hypotheses and discussion.

Best regards

Author Response

Best regards, dear members of the

Dear Publisher,

The revised manuscript of the original research article titled “Relationship between child abuse and delinquent behavior in male adolescents deprived of liberty” (Manuscript ID: ijerph-2016895) is attached and has been revised and corrected following the valuable comments of the reviewers.

Review #1

Mentions:

a.-  I advise you to look at other articles to improve yours.

The reviewer's suggestions were followed and as my tutors, Dr. Eloy Girela Lopez and Dr. Cristina Beltran, published an article entitled "Study of the psychometric properties of the Spanish version of the Measure of Moral Distress for Health Care Professionals (MMD-HP-SPA)” (ID: ijerph-2021665)” in this journal, under their recommendations, the indicated improvements were made in order to comply with the regulations of Magazine

b.-The introduction is a bit messy and some content is not related to the article (eating disorders).

The introduction was improved because, due to an involuntary error when collecting the information, literature on eating disorders left, which have already been eliminated from the manuscript.

c.- There are also formatting errors and missing references. In the methodology, sections are missing (or appear at the end of the article).

Corrections were made in relation to the bibliographical references and the methodology was organized since the sections as mentioned by the reviewer are present, but they needed to be numbered to appreciate the sections.

d.- The objectives and hypotheses should be further developed. and finally, the results should be linked to the objectives, hypotheses and discussion.

The objectives and hypotheses were developed within the legal and ethical framework established by the Ecuadorian state and international regulations for the study of these cases.

The results were broken down linking objectives, hypotheses and discussion

Eternal gratitude to the editor and reviewer for the observations made since they allowed us to improve the manuscript.

Reviewer 2 Report

Revisions:

Since this is a very impressive and unique manuscript, the contents contain significant concepts that its reading audience will appreciate. Therefore, I do not have any suggestions for revision.

Author Response

Best regards, dear members of the

Dear Publisher,

The revised manuscript of the original research article titled “Relationship between child abuse and delinquent behavior in male adolescents deprived of liberty” (Manuscript ID: ijerph-2016895) is attached and has been revised and corrected following the valuable comments of the reviewers.

Review # 2

Mentions:

Since this is a very impressive and unique manuscript, the contents contain significant concepts that its reading audience will appreciate. Therefore, I do not have any suggestions for revisión

Greetings dear colleague

I tell you that carrying out investigative field work with minors is very complex, now imagine the legal, legal, ethical difficulties, when carrying out work with minors who violate the law, where they are protected by the Constitution of the Ecuadorian state, the Childhood and Adolescence Code and even international rights.

Eternal gratitude for your analysis and reflection on this manuscript.

Reviewer 3 Report

This is a well structured and well written study with careful attention to the topic.  I was impressed that after making notes on shortcomings, and reading the referenced material, I then read the limitations section - I found all of my concerns and opportunities were identified precisely.  I personally find the 5-point Likert scales rather clunky and the results are therefore a little simplistic.

Given the access to so much ancillary information (from the files and also in interviews) it might have been possible to go beyond what is presented here, though agree the real issues lie in causality, which requires more time depth - but that points to another opportunity - what are the long term consequences as the offenders age - some of the references do suggest this is a fruitful objective for follow-up studies if protocols allow this. There are many questions, such as 'what are the implications of multi-generational persistence of abuse? or multiple offending?'  Another point, if this is repeated for women, would be to consider whether the justice system treats men and women equitably.

A few cosmetic things would vastly improve the paper for readers.  These should be simple to fix.

Minor typos: e.g. repetition of "neglect" in line 27, frequent retention of the Spanish "y" instead of "and" e.g. line 68, "Referencias" for "references" in line 371.

References need to be consistently and properly formatted (looks very rushed - mixed fonts, mixed case) - also the UNICEF and WHO references (as examples) are also published in English. For an English-speaking audience, the English version would be a better option for citation. The Boisvert paper was first on-line in 2018 but the cited version is 2019. The weblink to the entry "Censuses, NI 2019" did not work for me. However, a very good and useful set of references.

Tables: for the same reason suggest changing DE to the English SD (table 2 for example has DE in the table and SD in the footnote). Tables should be checked for formatting too - Table 5 should have "Fringe Context" on a new row and centred.

Author Response

Best regards, dear members of the

Dear Publisher,

The revised manuscript of the original research article titled “Relationship between child abuse and delinquent behavior in male adolescents deprived of liberty” (Manuscript ID: ijerph-2016895) is attached and has been revised and corrected following the valuable comments of the reviewers.

Review # 3

Mentions:

a.- This is a well structured and well written study with careful attention to the topic.  I was impressed that after making notes on shortcomings, and reading the referenced material, I then read the limitations section - I found all of my concerns and opportunities were identified precisely.  I personally find the 5-point Likert scales rather clunky and the results are therefore a little simplistic.

R.- The Likert scale on 5 points was used, being 1 (never) and 5 (always) to analyze the types of abuse, due to the legal, legal and ethical limitations of the subject in being able to extract more information from minors, since protected by national and international legal regulations

b.- Given the access to so much ancillary information (from the files and also in interviews) it might have been possible to go beyond what is presented here, though agree the real issues lie in causality, which requires more time depth - but that points to another opportunity - what are the long term consequences as the offenders age - some of the references do suggest this is a fruitful objective for follow-up studies if protocols allow this. There are many questions, such as 'what are the implications of multi-generational persistence of abuse? or multiple offending?'  Another point, if this is repeated for women, would be to consider whether the justice system treats men and women equitably.

R.- As the reviewer mentions, the subject is projected to a Psycho-Emotional - Psychosocial future of Young offenders of the law to their adulthood, but the National legal regulations together with the Internment legal framework protect the minor and his information. Reason why under your suggestion it was considered important to point out for the reader these difficulties found in the constitution, Childhood / Adolescence Code and in the Comprehensive Organic Criminal Code (COIP)

c.- A few cosmetic things would vastly improve the paper for readers.  These should be simple to fix.

Minor typos: e.g. repetition of "neglect" in line 27, frequent retention of the Spanish "y" instead of "and" e.g. line 68, "Referencias" for "references" in line 371.

R.- These typographical errors that inadvertently appear in the manuscript have been corrected

D.- References need to be consistently and properly formatted (looks very rushed - mixed fonts, mixed case) - also the UNICEF and WHO references (as examples) are also published in English. For an English-speaking audience, the English version would be a better option for citation. The Boisvert paper was first on-line in 2018 but the cited version is 2019. The weblink to the entry "Censuses, NI 2019" did not work for me. However, a very good and useful set of references.

R.- The restructuring and updating of the bibliographic references was carried out according to the observations of the reviewer and the regulations of the journal.

E.- Tables: for the same reason suggest changing DE to the English SD (table 2 for example has DE in the table and SD in the footnote). Tables should be checked for formatting too - Table 5 should have "Fringe Context" on a new row and centred.

R.- Corrections were made according to the reviewer's instructions.

Eternal gratitude for the observations made, as this allowed the improvement of the manuscript, which i am sure with your healthy criticism, readers will have a real conceptual framework on this research subject in young offenders of the law

Round 2

Reviewer 1 Report

Dear authors,

I hope you are well.

I have reviewed the manuscript and there are changes in the article. However, it is not yet ready for publication. I encouraged you to review other articles and you have replied that you have reviewed them. But it strikes me that the methodology is at the end of the paper.

The current version is not ready for publication.

The order should be:

- introduction

- Methodology

- Results

- discussion and conclusions

- references

Author Response

Best regards, dear editor and reviewer

Dear

Please, find enclosed the revised manuscript of the original research article entitled “Relationship between child abuse and delinquent behavior in male adolescents deprived of liberty” (ID: ijerph-20216895) which we have revised and corrected following the valuable comments of the reviewers.

Referee 1:

We proceeded to order the manuscript according to your indications, we thank you for your sound criticism as it allows us to comply with the regulations for its publication.

Hugs and blessings.

Sincerely

Patricio A. Vallejo Valdivieso 
